ecology/behaviour

sediment, biological interactions, mutualisms, damselfishes, coral, fish–coral interactions

**Author for correspondence:**
T. J. Chase
e-mail: tory.chase@my.jcu.edu.au

# Damselfishes alleviate the impacts of sediments on host corals

T. J. Chase[1,2], M. S. Pratchett[2], M. J. McWilliam[2,3], M. Y. Hein[1,2], S. B. Tebbett[1,2] and M. O. Hoogenboom[1,2]

[1]Marine Biology and Aquaculture Group, College of Science and Engineering, and [2]ARC Centre of Excellence for Coral Reef Studies, James Cook University, Townsville, Queensland, 4811, Australia
[3]Hawai'i Institute of Marine Biology, University of Hawai'i at Manoa, Kaneohe, HI, 96744, USA

TJC, 0000-0002-2044-136X; MSP, 0000-0002-1862-8459;
MYH, 0000-0001-6951-4075; SBT, 0000-0002-9372-7617;
MOH, 0000-0003-3709-6344

Mutualisms play a critical role in ecological communities; however, the importance and prevalence of mutualistic associations can be modified by external stressors. On coral reefs, elevated sediment deposition can be a major stressor reducing the health of corals and reef resilience. Here, we investigated the influence of severe sedimentation on the mutualistic relationship between small damselfishes (*Pomacentrus moluccensis* and *Dascyllus aruanus*) and their coral host (*Pocillopora damicornis*). In an aquarium experiment, corals were exposed to sedimentation rates of approximately 100 mg cm$^{-2}$ d$^{-1}$, with and without fishes present, to test whether: (i) fishes influence the accumulation of sediments on coral hosts, and (ii) fishes moderate partial colony mortality and/or coral tissue condition. Colonies with fishes accumulated much less sediment compared with colonies without fishes, and this effect was strongest for colonies with *D. aruanus* (fivefold less sediment than controls) as opposed to *P. moluccensis* (twofold less sediment than controls). Colonies with symbiont fishes also had up to 10-fold less sediment-induced partial mortality, as well as higher chlorophyll and protein concentrations. These results demonstrate that fish mutualisms vary in the strength of their benefits, and indicate that some mutualistic or facilitative interactions might become more important for species health and resilience at high-stress levels.

# 1. Introduction

Positive species interactions play a critical role in community assembly, species coexistence and ecosystem function [1–3]. Mutualistic and facilitative relationships range from tightly coevolved symbioses (e.g. lichens, legumes and zooxanthellate

corals) to looser associations whereby certain taxa derive benefit from others in close proximity (e.g. plants–pollinators, and clownfish–sea anemones), both forming critical components of community interaction networks [2]. Many positive interactions arise from the ability of species to modify the local environment through nutrient enrichment or habitat modification, and therefore ameliorate stress for the benefit of their neighbours [4]. Studies from a range of systems demonstrate that the role of positive interactions increases under high-stress conditions [5–7], and interaction networks may shift to a 'survival mode' with a greater reliance on mutualism and facilitation. A major challenge, therefore, is to understand how positive interactions are likely to fare in the face of global environmental change, and how they might help communities deal with these stressors.

Coral reefs are hotspots of mutualistic and facilitative interactions [8–11]. Reef-building corals, for example, foster numerous interactions with obligate coral-dwelling invertebrates (e.g. *Trapezia* spp. crabs and other cryptofauna) and associated fish species that use corals for habitat or temporary refuge. Many of these interactions are mutualistic; augmenting the growth and overall health of their coral hosts [12–15]. Aggregative damselfishes, such as *Chromis* spp., and *Dascyllus* spp., provide beneficial services to corals, including increases in coral growth rates by up to 40% [16], reductions in black-band disease progression [17,18], subsidies of nitrogen and phosphorus, and increases in colony aeration by 60% [19,20]. Thus, although many studies highlight the breakdown of coral reef mutualisms during extreme stress [21], it is also possible that positive interactions could enhance system resilience by moderating effects of stressors on reef organisms [22–24].

Inputs of sediment to coastal environments and coral reefs have increased rapidly in recent times due to altered land-use practices [25,26], coastal development [27,28] and dredging [29,30]. It is widely acknowledged that high sediment levels can erode coral reef resilience via lethal and sublethal impacts on reef organisms [31–33]. For example, sediment reduces light levels, damages coral tissue, smothers polyps and reduces coral growth [34–37]. Furthermore, corals under high sediment levels are physiologically stressed [37,38], with reduced heterotrophy, the death of symbiotic *Symbiodiniacea* spp. [39], and the production of excess mucus to remove sediment [37,40–42]. While profound effects of sediment on the coral holobiont are evident, the potential for fish-derived benefits to assist corals stressed by sediments remains relatively unexplored. Indeed, behaviours of symbiont damselfishes such as 'water stirring' within colony branches and nocturnal aeration of stagnant inner colony areas [15] suggest that mutualistic associations may greatly enhance the capacity of host corals to withstand sediment stress.

The objective of this study was to test whether coral-dwelling damselfishes can alleviate the deleterious effects of sediment stress on their host coral colonies, by (i) reducing the accumulation of sediments within-host colonies; and/or (ii) moderating physiological damage, localized tissue loss and partial colony mortality. We hypothesized that fish movement and fish-derived services (i.e. 'water stirring' and nutrient subsidy) would assist corals under long-term, severe sediment stress (e.g. during sediment deposition following sustained dredging activity, storms or natural deposition by parrotfishes) through sediment removal, and that the varying behaviours (e.g. roosting position and colony visits [43]) of different damselfish species would benefit host corals to different extents. To assess this, a laboratory-based experiment was used to examine multiple physiological responses of corals to chronic sedimentation while hosting or not hosting aggregative damselfishes. Understanding the impacts of sedimentation on coral colonies within the context of coral–fish associations will provide new insights into the importance of mutualistic associations and their contributions to resilience in an increasingly modified environment.

# 2. Material and methods

## 2.1. Study site and specimen collection

Field sampling and the aquarium experiment were conducted between April and June 2017 on Orpheus Island, an inner-shelf, continental island of the Great Barrier Reef (GBR). Orpheus Island is located approximately 20 km from the Queensland coast and close to the Herbert (approx. 20 km) and Burdekin rivers (approx. 150 km) where seasonal flood plumes, storms, agricultural runoff and high-levels of resuspension often deposit sediment onto coral reefs [44]. Colonies of *Pocillopora damicornis* (averaging 13.5 cm in diameter) were collected from sheltered sites around the Palm Islands. *P. damicornis* is widely distributed on inshore and offshore reefs of the GBR, and exhibits high levels of occupancy by coral-dwelling damselfishes (Pomacentridae) [45,46]. Two damselfish species,

*Dascyllus aruanus* and *Pomacentrus moluccensis*, were collected from nearby sheltered reef areas using a weak solution of clove oil [47,48] and hand nets. These two damselfish species are common on the GBR and exhibit high levels of coral occupancy [45,46]. Fishes and corals were transported to the research station and transferred to 25 l flow-through seawater tanks. Corals and fishes were then allowed to acclimate to aquaria conditions for one week. All fishes were subjected to a brief freshwater rinse to remove contaminants [49] and weighed (wet weight, Kern PCB, John Morris Scientific balance, precision 0.001 g) to determine treatment group biomass. Resident coral cryptofauna (i.e. *Trapezia* spp. crabs and *Alpheus* spp. shrimp) remained within their host colonies to simulate a natural coral holobiont system; coral colonies were haphazardly assigned to different treatments so that any influence of these resident cryptofauna, and/or any other variability among individual coral colonies, did not drive differences among sediment and fish treatments.

## 2.2. Aquarium sediment deposition experiment

To test whether coral-dwelling damselfishes reduce the accumulation of sediments within occupied colonies, and thereby moderate deleterious effects of sediment on corals, 72 coral colonies were collected and haphazardly assigned to one of six treatments: (i) no sediment, no fish; (ii) no sediment with *P. moluccensis*, (iii) no sediment with *D. aruanus*, (iv) sediment added with no fish, (v) sediment added with *P. moluccensis*, and (vi) sediment added with *D. aruanus.* Fish treatments contained four individual damselfish from either of the two fish species, with biomass representative of colonies naturally found in the field [24,50]. *D. aruanus* ranged in size from 20 to 70 mm and weighed from 0.5 to 10.3 g, with an average group biomass of $11.9 \pm 0.3$ g. *P. moluccensis* ranged in size from 17 to 59 mm and weighed from 0.3 to 5.7 g, with an average group biomass of $8.5 \pm 0.6$ g. Diurnal (13.00–16.00) and nocturnal (20.00–22.00) fishes' behaviours in experimental aquaria were observed four to five times for each fish in each coral colony ($n = 24$ colonies with 96 fish per fish treatment, per time period), during the course of the experiment. Swimming positions of all *D. aruanus* or *P. moluccensis* in each replicate aquarium were recorded during spot checks ($n = 5$ diurnal checks, per colony, and $n = 4$ nocturnal spot checks per colony, each spread out over the course of the experiment), where the observer did not interfere with the fishes' behaviours. Nocturnal spot checks used a white light torch for illumination—each colony was illuminated for less than 10 s and did not induce movement by any of the resident fishes (see [43] for similar methods). Positional categories included: 'in colony branches' (within branching structure), 'outside colony' (vertically on top or to the side of colony) and 'under' (under colony structure).

Corals and fishes were maintained in outdoor aquaria (25 l volume), that received an inflow of new ambient filtered seawater (approx. $15 \, \text{l h}^{-1}$, re-circulating slowly enough to prevent sediment disruption). Aquaria were also fitted with an air-stone to maintain oxygen saturation of the water, but with sufficiently low air-flow rates to avoid disrupting sediments. Corals and fishes were fed daily to satiation with enriched *Artemia salina* nauplii. One coral fragment per colony, approximately 5 cm in length, was selected at random from the top planar surface of each colony, during acclimation (prior to adding fish or sediment) and again after 28 days of treatment exposure. Fragments ($n = 144$) were subsequently frozen in liquid nitrogen, transported to James Cook University, and coral tissues were analysed for chlorophyll density, protein density and tissue biomass [24]. Additional measurements of partial mortality were quantified from photos taken from above the coral at the beginning of the experiment and again after 28 days, after all sediment was removed. The two-dimensional area of the bleached or dead coral tissue was measured using ImageJ software [13,51,52].

A dose of 14 g of sediment was added to each tank using a funnel to spread the sediment evenly over the coral surface, daily for 28 days. This equated to standardized sedimentation rates of approximately $100 \, \text{mg cm}^{-2} \, \text{d}^{-1}$. We chose $100 \, \text{mg cm}^{-2} \, \text{d}^{-1}$ because this was similar to sedimentation rates quantified in the field around the Palm Islands (average approx. $140 \, \text{mg cm}^{-2} \, \text{d}^{-1}$, see electronic supplementary material, text S1, figures S1–S3 and table S1 for specifications on design and deployment). In addition, a level of $100 \, \text{mg cm}^{-2} \, \text{d}^{-1}$ was chosen to facilitate comparison with previous research that has explored the impacts of sediment deposition on corals under deposition rates ranging from 0.5 to $600 \, \text{mg cm}^{-2} \, \text{d}^{-1}$ in natural and controlled *ex situ* aquaria conditions [13,33,52–54]. While maximum sedimentation rates of greater than $100 \, \text{mg cm}^{-2} \, \text{d}^{-1}$ have been reported from around Magnetic Island, near Orpheus Island on the inner-shelf of the GBR [55], sedimentation rates of $100 \, \text{mg cm}^{-2} \, \text{d}^{-1}$ over prolonged periods are generally considered severe within the context of coral reef ecosystems [33,53]. However, many published studies have relied on sediment trap data to set their experimental treatments, and while this method is commonly used to quantify sediment accumulation rates it can

both overestimate and underestimate how much sediment is actually deposited on natural benthos [56,57]). However, the sedimentation rate used herein ($100\,\text{mg}\,\text{cm}^{-2}\,\text{d}^{-1}$) should be viewed as severe sediment deposition equivalent to levels occurring during dredging activities, wave-driven resuspension during tropical storms, and/or through direct deposition of sediments on coral colonies by parrotfishes [53,58,59]. Added sediment consisted of a combination of silicate, carbonate and organic particulates with grain sizes between 63 and 4000 µm, in a ratio of 4 (carbonate sediment, 63 µm) : 1 (siliciclastic sediment, 63 µm) : 2 (90–355 µm) : 3 (355–1400 µm) : 1 (1400–4000 µm), which is consistent with settled inshore sediments around the Palm Islands (see [60,61] for justification of size classes and sedimentation rates). Sediments were collected from local reefs, dried at 60°C for more than 24 h and sieved into size classes prior to experimental use (see [62–64] and electronic supplementary material, table S2 for sediment description and composition). Air-stone and water flow were turned off directly before sediment addition and remained off for 1 h to enable sediment settlement.

Sediments were carefully removed from the bottom of the tank every 3–4 days to mimic natural substrate clearing and to prevent any anoxic microbial build-up in experimental tanks (see electronic supplementary material, text S2, table S3, figure S4). To determine the amount of sediment remaining on the coral at the end of the experiment, each coral was carefully removed from its aquarium, placed into a labelled container full of seawater and shaken until all sediment was removed from the colony. Sediments were allowed to settle for more than 6 h in temporary collection buckets, transferred into labelled containers and transported to James Cook University for further processing. All collected sediments were rinsed with fresh water three times to remove salts, dried at 60°C (Axyos Microdigital Incubator) for more than 4 days, weighed for constant weight (g), sieved into three factions [65]: less than 125 µm (very fine sand and silt), 125–500 µm (fine to medium sand), 500–4000 µm (coarse sand to gravel) and weighed (using Kern PCB, John Morris Scientific balance, precision 0.001 g).

## 2.3. Data analysis

Variation in the total sediment load remaining on *P. damicornis* colonies after 28 days in aquaria was examined using a log-transformed linear model. In the model, 'fish presence' (no fish, *P. moluccensis*, *D. aruanus*) was treated as a fixed factor and only 'sediment added' treatment colonies were included in the analysis, as all colonies in the 'no sediment' treatment exhibited very low (less than 0.3 g) sediment accumulation during the experiment. Tukey's HSD comparisons were employed *post hoc* to assess differences among factor levels. Model fit was assessed using residual plots, all of which were satisfactory (normal and homogeneous). To assess whether the grain size distribution of sediments remaining on *P. damicornis* colonies differed among treatments, a permutational multivariate analysis of variance (PERMANOVA) was used. The PERMANOVA was based on a Euclidean distance matrix of standardized data, and once again fish presence was treated as a fixed factor. Pair-wise tests were used to determine where between factor level differences occurred. Homogeneity of dispersions for the PERMANOVA was tested using a permutation analysis of multivariate dispersions (PERMDISP). A canonical analysis of principle components (CAP) was employed following the PERMANOVA to visualize significant groupings, although grain size distributions were better visualized as bar graphs (see electronic supplementary material, figure S5).

Partial colony mortality of host *P. damicornis* colonies was analysed using a beta regression model with sediment and fish as interacting fixed factors. Due to the proportional nature of the data, the beta-binomial distribution with a logit-link was the most appropriate [66]. However, as this distribution is bounded between 0 and 1, a small constant (0.001) was added across the dataset. Model fit was assessed using residual plots, as above. Following the beta regression model, treatment comparison differences were assessed using least-square means (lsmeans) multiple comparisons *post hoc* (with a Tukey's correction).

Differences in coral tissue components (total chlorophyll, proteins and tissue biomass) were examined using two-way ANOVAs with sediment and fish treatments initially fitted as interacting fixed factors. Coral tissue components data were log-transformed. Tukey's HSD *post hoc* comparisons were used to examine between treatment differences. When interaction terms were not significant, additive models (sediment treatment + fish treatment) were performed. Model fit was assessed using residual plots, all of which were satisfactory (normal and homogeneous). Tissue components at the start of the experiment and after 28 days (end) were analysed separately, as all tissue component comparisons at the beginning were not significantly different.

Pearson's $\chi^2$ goodness-of-fit tests ($\chi^2$) were used separately for each damselfish species to determine non-random variation in diurnal and nocturnal fish position around host coral colonies in aquaria.

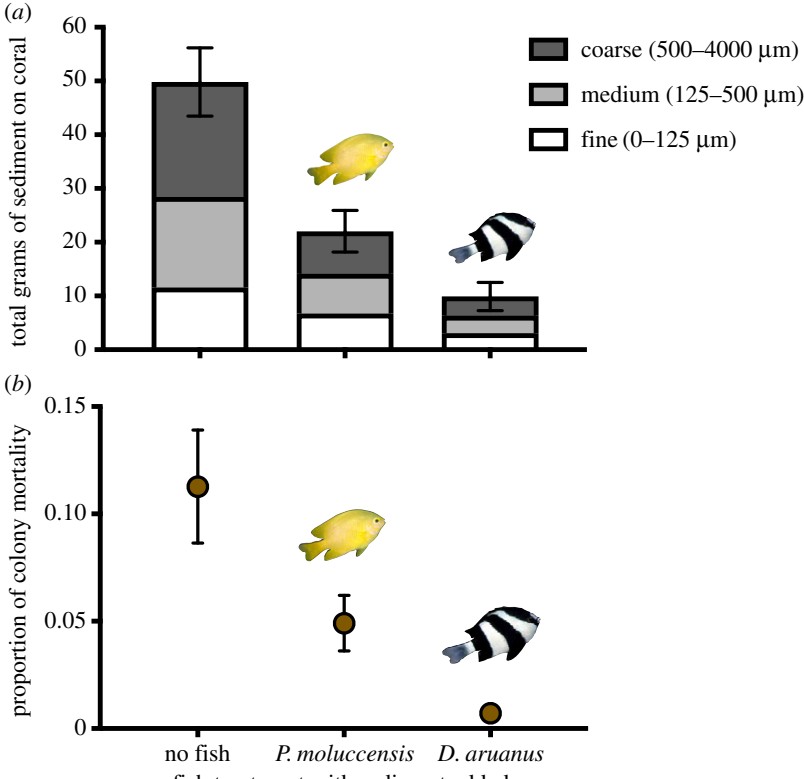

**Figure 1.** (a) Remaining sediment on P. damicornis colonies (approx. 13.5 cm diameter) after 28 days of approximately 14 g of sediment deposition in experimental aquaria sediment added corals. $n = 12$ coral treatments (except corals with sediment added, in which a colony died on day 25 and was removed from analysis). Treatments included colonies with different fishes (no fish, three P. moluccensis and three D. aruanus) and sediment (no sediment and with sediment added at a rate of approximately 100 mg cm$^{-2}$ d$^{-1}$ for 28 days). Error bars show s.e. As values for no sediment treatments were very low (less than 0.29 g), they were not included in this figure. Bar colours represent grain size fractions as follows: dark grey is coarse (500–4000 μm), grey is medium (125–500 μm) and white is fine (0–125 μm) sediment. (b) Average levels of whole P. damicornis colony partial mortality, measured after 28 days of experimental fish and sediment treatments.

Diurnal and nocturnal positions were the count of multiple observations (the sum of $n = 5$ and $n = 4$ observations, respectively, treated as replicates rather than repetitive time points). This was deemed appropriate for the categorical nature of the spatial position data. All analysis was performed in the statistical software R [67] using the *betareg* [66], *multcomp* [68] and *lsmeans* [69] packages. Multivariate analysis was performed in PRIMER 7.0 PERMANOVA+.

# 3. Results

## 3.1. Fishes' removal of sediment in aquaria

The total weight of accumulated sediment on host corals varied according to the presence or absence of specific host fishes ($F_{2,33} = 28.22$, $p < 0.001$, figure 1a; electronic supplementary material, table S3). Sediment commonly pooled on the upper horizontal surfaces of the coral colonies, with the majority becoming trapped within branch connection points. Sediment treatment colonies of *P. damicornis* hosting *D. aruanus* exhibited the lowest levels of accumulated sediment (approx. $10 ± 2.6$ g), which was twofold less than sediment treatment colonies hosting *P. moluccensis* (approx. $22 ± 3.9$ g, Tukey's HSD: $p = 0.002$), and nearly fivefold less than vacant sediment treatment colonies (approx. $49 ± 6.3$ g, Tukey's HSD: *D. aruanus*, $p < 0.001$; *P. moluccensis*, $p = 0.002$; see electronic supplementary material, table S4). Sediment grain size fractions left on *P. damicornis* colonies after 28 days varied by treatment (PERMANOVA: pseudo-$F_{2,33} = 3.0615$, p(perm) = 0.0485; electronic supplementary material, table S5 for homogeneity of sediment grain size distributions), with higher amounts of medium and coarse sediments removed from fish treatment colonies. Pairwise tests revealed that grain size fractions of

sediment remaining on colonies were significantly different between fish-absent colonies and *D. aruanus* present colonies (pairwise test, $t = 2.061$, p(perm) = 0.041) and *P. moluccensis* present colonies (pairwise test, $t = 2.177$, $p = 0.028$). However, grain size fractions on colonies did not differ between colonies with *D. aruanus* and *P. moluccensis* (pairwise test, $t = 1.304$, p(perm) = 0.2095). Sediments remaining on non-sediment treatment colonies were very low (less than 0.29 g) for all three treatments and were probably a result of residual sediments within the aquarium system.

Partial colony mortality was explained by the presence or absence of fish under sediment stress (electronic supplementary material, figure S6). Sediment-free colonies of *P. damicornis* did not exhibit any signs of partial mortality, and colonies subjected to daily sediment treatments exhibited an average of 5.6% partial mortality over the course of 28 days, ranging from less than 1% to 32% (figure 1*b*). Areas of partial mortality were usually limited to the site where sediments directly settled, and generally, there were no visible impacts on the healthy coral tissue that was located less than 1 cm away from the impacted tissue. The highest average partial mortality (11.2% ± 0.03) was observed in the sediment addition with no fish treatment (figure 1*b*), which was twofold higher than the partial mortality of colonies with *P. moluccensis* (4.9% ± 0.01), (lsmeans: (no fish versus *P. moluccensis*) $p < 0.05$), and fourfold more than colonies with *D. aruanus* (lsmeans: (no fish versus *D. aruanus*) $p < 0.001$). Host colonies with sediment added and *D. aruanus* exhibited very low partial mortality (less than 1%). Indeed, partial colony mortality on sediment-added colonies with *D. aruanus* was not significantly different from that of sediment-free colonies (electronic supplementary material, tables S6 and S7).

## 3.2. Impacts of sediment and fishes on coral tissues

Prior to sediment and fish treatments, chlorophyll density ($\bar{x} = 5.4 \pm 0.4$ µg cm$^{-2}$), protein concentration ($\bar{x} = 1.8 \pm 0.5$ mg cm$^{-2}$) and tissue biomass ($\bar{x} = 1.9 \pm 0.0$ mg cm$^{-2}$) were not significantly different among treatments (ANOVA, total chlorophyll (sediment*fish): $F_{2,59} = 0.165$, $p = 0.849$), total protein (sediment*fish): $F_{2,66} = 1.486$, $p = 0.234$; tissue biomass (sediment*fish): $F_{2,66} = 1.244$, $p = 0.295$) ($p > 0.05$ for all other factors for the three tissue components, see electronic supplementary material, table S8). After 28 days of sediment and fish treatments, there were reductions in coral tissue components in sediment-added colonies with no damselfish (figure 2).

Overall, corals exposed to sediments and hosting *D. aruanus* exhibited the lowest coral tissue stress. Specifically, chlorophyll levels in colonies stressed by sediments and hosting *D. aruanus* were twofold higher (7.37 ± 1.18 µg cm$^{-2}$) compared with colonies stressed with sediment but not hosting fish (3.24 ± 0.59 µg cm$^{-2}$), which was statistically significant (Tukey's HSD *post hoc*: $p < 0.05$). By contrast to *D. aruanus*, *P. moluccensis* had no significant effect on chlorophyll levels (table 1; electronic supplementary material, table S8). The interaction between sediments and fish treatment was not significant for chlorophyll (ANOVA: $F_{2,61} = 1.216$, $p = 0.304$).

Patterns in total protein concentration at the end of the experiment were similar to those for total chlorophyll in that colonies hosting *D. aruanus* had the highest total protein levels (figure 2*b*). Despite the higher protein levels in colonies hosting *D. aruanus*, the only statistically significant difference occurred between colonies with no sediment added and hosting *D. aruanus* (2.22 ± 0.2 mg cm$^{-2}$) and colonies with sediment added, but with no fish (1.25 ± 0.2 mg cm$^{-2}$, Tukey's HSD *post hoc*: $p < 0.01$; figure 2*b*, table 1). Again, the interaction between sediments and fish treatment for protein content was not significant (ANOVA: $F_{2,65} = 2.682$, $p = 0.076$). Finally, no significant differences in tissue biomass were detected among treatments at the end of the experiment (ANOVA: (fish effect) $F_{2,65} = 2.631$, $p = 0.079$; table 1 and figure 2*c*; electronic supplementary material, table S8).

## 3.3. Spatial position of damselfishes in aquaria

Diurnal and nocturnal positions differed between *D. aruanus* and *P. moluccensis* (figure 3). During the day *D. aruanus* swam less than 80% of its time outside the colony branches, mainly on top of the colony ($\chi^2 = 174$, d.f. = 2, $p < 0.001$). By contrast, *P. moluccensis* spent most of its time within the branches or under the colony ($\chi^2 = 69$, d.f. = 2, $p < 0.001$). However, at night, *D. aruanus* preferentially slept within host colony branches ($\chi^2 = 469$, d.f. = 2, $p < 0.001$), while *P. moluccensis* was less specific about roosting locations, spending nearly equal time in the colony, outside the colony or under the colony ($\chi^2 = 1.2$, d.f. = 2, $p = 0.56$).

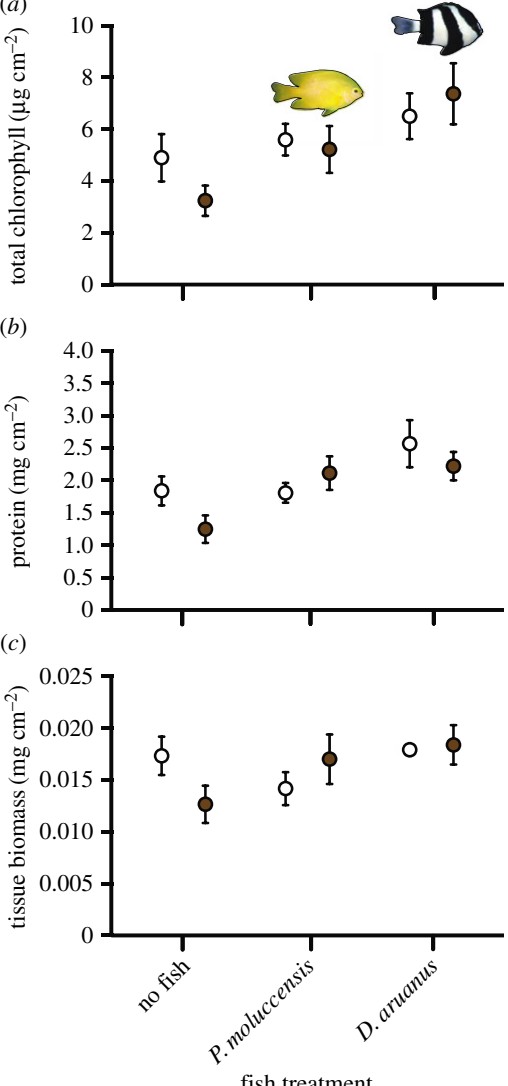

**Figure 2.** Sample fragment tissue compositions at the end of 28 days of experimental sediment and fish treatments: (*a*) total chlorophyll (chl *a* + chl *c*, $\mu$g cm$^{-2}$), (*b*) protein (mg cm$^{-2}$), (*c*) tissue biomass (ash-free dry weight, mg) for *P. damicornis* colonies in experimental aquaria with different fishes (no fish, three *P. moluccensis* and three *D. aruanus*) and sediment treatments (no sediment: white dots and with sediment added at a rate of approx. 100 mg cm$^{-2}$ d$^{-1}$: grey dots for 28 days). Error bars show s.e. Refer to table 1 and electronic supplementary material, table S8 for comparisons among treatments.

**Table 1.** Tukey's HSD *post hoc* for multiple comparisons of tissue components (total chlorophyll, total protein and tissue biomass) from two-way additive ANOVAs (sediment treatment + fish treatment) and *p*-values. Significant *p*-values are in italics.

| tissue component | comparison | *p*-value |
|---|---|---|
| total chlorophyll | *P. moluccensis* − *D. aruanus* | 0.5154 |
| | *P. moluccensis* − no fish | 0.1492 |
| | *D. aruanus* − no fish | *0.0117* |
| total protein | *P. moluccensis* − *D. aruanus* | 0.1686 |
| | *P. moluccensis* − no fish | 0.2667 |
| | *D. aruanus* − no fish | *0.0063* |
| tissue biomass | *P. moluccensis* − *D. aruanus* | 0.4217 |
| | *P. moluccensis* − no fish | 0.9128 |
| | *D. aruanus* − no fish | 0.2263 |

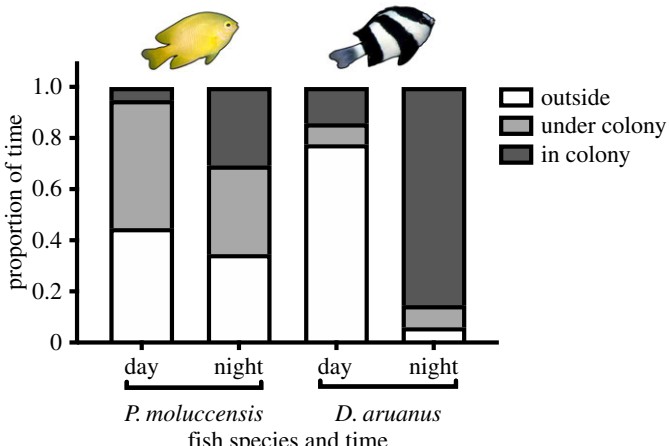

**Figure 3.** Average nocturnal (approx. 21.00) position (proportion ± s.e.) of *P. moluccensis* and *D. aruanus* in relation to small (approx. 13.5 cm diameter) *P. damicornis* colonies in experimental aquaria (25 l cylindrical tanks) at Orpheus Island Research Station. Half of the total coral colonies (*n* = 72) were exposed to sediment treatments. Fish treatment and numbers: *n* = 72 *D. aruanus* on 24 colonies and *n* = 72 *P. moluccensis* on 24 colonies.

# 4. Discussion

This study demonstrates that the presence of coral-dwelling fishes reduces accumulation of sediment on host corals during high sedimentation conditions, and thereby moderates the localized tissue damage caused by sediments. Moreover, colonies with fishes had higher chlorophyll and protein concentrations compared with unoccupied corals when subjected to severe sediment stress. These results suggest that coral-dwelling damselfishes and, potentially, other cryptofauna provide a 'housekeeping service' to branching corals, adding to the growing list of recognized indirect and direct services that fishes provide to host corals [15,19,20,50]. *D. aruanus*, in particular, had strong mutualistic effects on its coral host, as high sediment-exposed coral colonies hosting *D. aruanus* had equivalent levels of partial mortality to coral colonies that were not exposed to any sediments. Consequently, fish presence can negate the negative impacts of severe sediment deposition (e.g. following direct deposition by parrotfishes [59,70], or heavy wave action) on coral physiology, potentially leading to higher fitness in corals with associated fishes due to larger energy reserves (i.e. nutrients and photosynthetic efficiency), increased overall growth [16,24,71,72], increased reproductive output [15] and enhanced colony resilience.

The removal of sediments from host corals has been demonstrated previously for coral-dwelling crabs (*Trapezia* spp.) and shrimps (*Alpheus* spp.), which can generate significant increases in coral growth in the field [13,52,73]. However, levels of sediment removal (or limited accumulation) by *D. aruanus* and *P. moluccensis* (recorded here) were much greater (approx. 95% of sediments removed) than those recorded for *Trapezia* spp. crabs (less than or equal 60%, as reported in [13]). While sediment removal by coral-dwelling *Trapezia* spp. crabs may be intentional [13], sediment removal by fishes may be more indirect and unintentional, caused primarily by their movements in and around the coral, but also via other mechanisms such as (i) additional coral mucus production through abrasion and impairment [73,74], (ii) enhanced coral polyp expansion and cilia movement [73,75], and (iii) inadvertent or passive removal due to capture of sediment in gills [76]. Active removal of sediment by damselfishes appears to be less frequent, but not uncommon. Indeed, we observed damselfishes deliberately removing sediment particles by picking them up in their mouths in this experiment (also seen previously in *D. marginatus* [15]), or blowing them off the coral to clear their preferred roosting areas and to tend to their habitat area [15,77]. Damselfishes also appear to be effective at clearing sediments around the base of coral colonies, excavating areas under the branches. This activity would allow for further coral expansion around coral attachment points and deter detrimental bacterial activity, anoxia (present in the sand [78,79]) or disease in coral colonies [30].

The effectiveness of damselfishes in moderating sediment deposition varied between the two focal damselfish species, which may be attributable to the strength and intensity of interactions between the fishes and *P. damicornis* colonies. For example, *D. aruanus* exhibited high levels of colony visits (potential water stirring behaviour and nutrient subsidy) and sleeps exclusively within its host colony

branches [43]. By contrast, *P. moluccensis* is less regular in its nocturnal roosting position and exhibits fewer colony visits (figure 3). Furthermore, the extent of sediment removal by fishes is probably greatest, and most beneficial, at the very beginning of sediment exposure, and at night when oxygen levels within the inner branches decline [80]. This is supported by the fact that during daily sediment doses and at night, *D. aruanus* retreat or roost within the branches of their colony, subsequently augmenting colony aeration and water flow [20].

This study suggests the importance of some mutualistic or facilitative interactions may become greater as abiotic stress levels increase, as seen in terrestrial systems [2,6,7]. Consequently, the positive net effect of hosting damselfishes on corals [2] is likely to be heavily context-dependent and may be particularly important for specific coral colonies on sheltered, inshore reefs, where negative impacts of nutrient-laden terrigenous sediments are the most pervasive [81,82]. This notion is supported by previous results which have highlighted that the positive impacts of aggregating damselfishes on coral growth are highest in sand patches and reef slope/base areas [50,83]. Moreover, *D. aruanus*, *P. moluccensis* and other coral-inhabiting damselfishes are most commonly found on corals located in sheltered (flow less than 21.2 cm s$^{-1}$, see [84,85]), reef/sand edge environments [83,86]. Sheltered sites with low hydrodynamic energy facilitate the settlement of finer sediments suspended in the water column [37,55], maximizing sedimentation rates that can lead to the smothering of corals, a common phenomenon on many inshore reefs [31]. As a result, there is spatial congruency between where the damselfishes with greater positive coral interactions are located and the strength of their benefits to host coral at the scale of habitats, as there are often branching corals on inshore sites [87–89]. Since removal of symbiont fishes lowers coral growth and reproduction rates [15], the loss of resident fishes will probably have a detrimental effect on coral colony health under high sedimentation, similar to bleaching conditions [24].

It should be noted that explicitly uncoupling the impacts of fish presence and/or cryptofauna presence (i.e. coral benefiting services) with sediment removal on coral health will require additional tests of the physical mechanisms in isolation. The interaction nature between damselfishes and different coral taxa, will certainly produce different levels of sediment removal due to colony morphology and damselfish use and abundance [43,45,90,91]. While the presence of resident cryptofauna has been demonstrated to impact the behaviour of corallivorous fishes and other predators [50,92], no impacts on resident damselfishes have been previously documented. Furthermore, as cryptofauna were standardized across experimental corals (natural cryptofauna left in corals and the coral colonies haphazardly allocated to treatments) to retain a natural coral holobiont, fish behaviour and impacts on coral health reported here are in addition to the natural coral holobiont processes. Removing such cryptofauna from coral colonies would reduce the ecological relevance of this study.

In some circumstances high levels of sediment deposition on reefs is associated with increased turbidity levels [55], and such high turbidity levels can also alter fishes' behaviours [93,94]. For example, reduced foraging distances exhibited by damselfishes in high water flow conditions [43,95] or turbid water [96,97], will alter the nature of damselfish–coral interactions, resulting in variable amounts of sediment removed during sedimentation events [16,43]. This is important as our method of sediment delivery, funnelling sediment onto corals, has a minimal effect on turbidity levels and associated fish behaviour. However, while high sediment deposition on corals is often associated with high turbidity [30,36], deposition of fine sediments is just one range of processes by which sediment can be deposited on corals. Consequently, conditions with high sedimentation but low turbidity can also occur on reefs. For example, high sediment deposition by parrotfishes can play a major role in sediment accumulation at local scales, independent of background turbidity levels [59,98–100]. Indeed, scraping parrotfishes, such as *Scarus rivulatus*, are abundant on inner-shelf reefs of the GBR [101,102], and therefore contribute substantially to sediment dynamics on these reefs [59,70,103]. Furthermore, scraping parrotfishes rework existing settled sediment (the composition and size of sediments used herein was based on settled sediments in the Palm Islands) rather than producing 'new' sediments through bioerosion like excavating parrotfishes [59,104]. As such, the sediments used herein are likely to reflect those that parrotfishes interact with and deposit on reefs around the Palm Islands.

The sediment levels used in the present study were designed to reflect severe, prolonged sediment deposition. As such, the levels used herein are higher than average background sedimentation levels that are often reported [55,57]. However, prior research has documented that select natural coral populations experience sediment deposition rates exceeding 200 mg cm$^{-2}$ d$^{-1}$ [33], which is considerably higher than the sedimentation levels used in the current experiment. Nevertheless, the severe sedimentation levels used herein are uncommon in coral reef ecosystems and our results revealing the positive effects of fishes on corals should be interpreted within this context. Our study represents a step forward in determining the nature of fish–coral interactions under sediment stress, and

highlights that a positive interaction can occur in certain circumstances. There is scope for future research to explore the relationship between fishes and corals under a more nuanced range of sedimentation levels, additional branching morphologies/taxa, and other non-visible and sublethal impacts.

The above point also highlights the ongoing knowledge gaps in our basic understanding of sediment dynamics on coral reefs (i.e. understanding links between turbidity levels, sedimentation rates and sediment accumulation on the benthos) [55,57,105] and how best to quantify these processes [56,106]. Indeed, while sediment traps have been the mainstay of measuring sedimentation for decades, a burgeoning body of evidence is highlighting that they do not measure sedimentation, but instead quantify a 'trapping rate' and only trap a subset of the sediment passing over coral reef ecosystems [56,57,107]. Sediment traps, therefore, may only provide a partial, and potentially biased, view of sediment dynamics on coral reefs. As such, we are far from a comprehensive understanding of sediment dynamics in coral reef ecosystems. Such an understanding will be critical if we are to accurately calibrate experiments to fully assess the impacts of sediments on the multitude of organisms that inhabit coral reefs, and the interactions between these organisms.

Increased sediment inputs are one of the main stressors driving the degradation of reefs [36,88,108]. The impacts of these sediments on coral reef ecosystems range from sublethal effects on individual coral colonies, to sediment-driven regime shifts altering the functioning of benthic communities [109]. This study demonstrates that small aggregating damselfishes can help the existing coral holobiont alleviate the negative effects of sediments deposited on corals under severe sediment deposition, by removing sediments and enhancing coral colony survival. Such benefits have the potential to act as stabilizing forces, facilitating the persistence and growth [16,110,111] of the coral holobiont (including endosymbionts and exosymbionts) in the face of anthropogenic and natural stressors [22,23]. These positive interactions link high diversity to high productivity under stressful environmental conditions [5], increasing survivorship of interacting species in the face of certain global climate change conditions. Unfortunately, mutualist damselfishes, those proposed to offer the greatest benefits to corals under high sediment stress, are also some of the most sensitive fishes to environmental changes [90]. As such, these important mutualisms may become less prevalent with ongoing reef degradation, limiting the propensity of fishes to support coral colony health when exposed to widespread environmental change. By developing a mechanistic understanding of the association between ecologically important aggregating damselfishes and their coral hosts, this study sheds new light on the manifestation of context-dependent symbioses in coral reef systems.

Ethics. This project was implemented in accordance with the Great Barrier Reef Marine Park Authority permit (G17/379187.1 and G15/38002.1), James Cook University Animal Ethics Permit (A2351) and James Cook University General Fisheries Permit (170251). All corals and damselfishes were returned to the site of collection (following JCU Ethics permit A2351), and select coral fragments (less than 8 cm in length) were sacrificed for further laboratory tissue analysis, per GBRMPA permit G17/379187.1 None of the corals or damselfishes collected were protected species.

Data accessibility. All data regarding sediments, coral mortality and tissue composition for *P. damicornis* corals in aquaria are available from the Dryad Digital Repository: https://datadryad.org/stash/share/TVDVmDFr1X0ijVL4aCdfru-k7mQyd4WjoryQVIIPjLo [112].

Authors' contributions. This paper has multiple authors and our individual contributions: T.J.C., M.S.P. and M.O.H. designed the study. T.J.C., M.J.M. and M.Y.H. conducted the fieldwork and data collection. T.J.C. and S.B.T. completed the data analysis and T.J.C. and M.O.H interpreted the data. T.J.C., M.S.P. and M.O.H, wrote the manuscript. All authors contributed critically to drafts, have read and approved the final manuscript.

Competing interests. We declare we have no competing interests.

Funding. This research was funded by the Australian Research Council to the ARCCOE for Coral Reef Studies CE140100020 and James Cook University. The funders have no role in study design, data collection and analysis, decision to publish, or preparation of the manuscript. This research was funded by James Cook University and the ARC Centre of Excellence for Coral Reefs Studies.

Acknowledgements. We thank the Orpheus Island Research Station staff, Dan Roberts, Molly Scott, Taylor Whitman, Alejandro Usobiaga, Allison Paley and Andrew Negri for their field support and technical assistance.

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
