## [Reviewer comments · Royal Society Open Science]

Review History

RSOS-190966.R0 (Original submission)

Review form: Reviewer 1

Is the manuscript scientifically sound in its present form?

Yes

Are the interpretations and conclusions justified by the results?

Yes

Is the language acceptable?

Yes

Is it clear how to access all supporting data?

Yes

Do you have any ethical concerns with this paper?

No

Have you any concerns about statistical analyses in this paper?

I do not feel qualified to assess the statistics

Recommendation?

Accept with minor revision (please list in comments)

Comments to the Author(s)

The manuscript by Chase et al. is a revision of a previously submitted manuscript. The authors have satisfactorily addressed the reviewer's concerns and in my opinion the manuscript is acceptable for publication following a few minor revisions that are listed below.

Line 83. There has been a revision of Symbiodinium systematics. Symbiodinium is now a genus within the family Symbiodiniaceae (LaJeunesse et al. 2018). The authors should use the new systematics and change Symbiodinium in the sentence to read Family Symbiodiniaceae.

Line 383: the authors state "at night when oxygen levels within the inner branches decline" without providing a citation. Please cite Shashar et al. (1993) for oxygen levels in corals at night.

Line 396. The sentence: "These conditions, therefore, maximise sedimentation rates that can lead to the smothering of corals, characteristic of many inshore reefs [31], leading to greater positive interactions between fishes and their coral hosts." This is an odd sentence that does not work. Please re-write. Perhaps re-work the "characteristic of many inshore reefs" somehow differently in the sentence.

Line 400. The sentence: "Removal of symbiont fishes, therefore, will likely lead to decreases in coral health under high sedimentation, similar to bleaching conditions [24]." Liberman et al. (1995) conducted a damselfish removal experiment and demonstrated that damselfish removal led to lower coral growth and reproductive rates. I think this citation would enhance the sentence. Perhaps change to:
Since removal of symbiont fishes lowers coral growth and reproduction rates (Liberman et al. 1995), fish removal will likely detrimentally affect coral health under high sedimentation, similar to bleaching conditions [24].

Figure 1: I like the fact that the authors have pictures of the fish in the figure. Since in the 'no sediment' treatment no sediment was added (a) and no coral mortality occurred (b), the left side of the figure with the 'no sediment' treatment is not needed in the figure and can be stated verbally. The figure looks odd with only half a figure. I suggest removing the 'no sediment' information from the graph and removing the 'sediment added' from the right hand portion of the graphs and stating that information in the legend. By doing so, Figure 1 will be more in line with Figure 2.

Figure 3: There is a mistake in the figure. The species name underneath the picture for *P. moluccensis* is actually *D. aruanus* and not *P. moluccensis*. Given that the rest of the graphs have *P. moluccensis* data appearing first, I assume that the error is in the species names that appear under the bars and not in the order of the bars.

Review form: Reviewer 2**Is the manuscript scientifically sound in its present form?**

No

Are the interpretations and conclusions justified by the results?

No

Is the language acceptable?

Yes

Is it clear how to access all supporting data?

Yes

Do you have any ethical concerns with this paper?

No

Have you any concerns about statistical analyses in this paper?

No

Recommendation?

Reject

Comments to the Author(s)

The authors have added text to address some of the more superficial comment of the reviews, but the fundamental issue of this work not justifying the conclusions drawn in this manuscript remain. There is still no meaningful ecological relevance to the study as completed, and the issue of interactive effects of cryptofauna and fishes remains - despite caveats added to the text. My conclusions from my initial review still stand. This work appears to be a result of a study conducted to show a predetermined result, and not a careful examination of a hypothesis developed from careful natural observations, or careful reading of the literature.

Review form: Reviewer 3

Is the manuscript scientifically sound in its present form?

No

Are the interpretations and conclusions justified by the results?

No

Is the language acceptable?

Yes

Is it clear how to access all supporting data?

Yes

Do you have any ethical concerns with this paper?

No

Have you any concerns about statistical analyses in this paper?

No

Recommendation?

Major revision is needed (please make suggestions in comments)

Comments to the Author(s)

The manuscript entitled 'Fishes alleviate the impacts of sediments on coral hosts' presents an interesting study on how two species of Damsel fish reduce coral tissue mortality and maintain coral health through the removal of sediments from the surface area of a branching coral. The manuscript is well written and easy to follow, and the accompanying data analysis is thorough

and well presented. However, I'm struggling to see value of these results given the lack of natural ecological relevance. This relates to the use of 100 mg.cm².day as the sedimentation rate, which the authors claim falls well within realistic values as measured on the reefs. My main points related to the use of this sedimentation rate are:

1. This rate is exceptionally high and rarely recorded on reefs except during big resuspension events or from dredging events.
2. This value have been measured using sediment traps which those working in these field know vastly over estimate sedimentation rates. One of the references that the authors use to justify the 100 mgcm²day - states this exact fact and does not back up their range of 30 to 360 mg cm² day - in fact the very opposite, and then goes on to make the point about lab experiments failing to use realistic values for these types of experiments
3. And even if 100 mg.cm².day sedimentation rate was a thing - to get to these sedimentation values, there would be exceptionally turbid waters for several days - which I suspect would also impair fish behaviour, which has stated at the end of the discussion, are sensitive to environmental change. Plus there are several studies (e.g. Wenger) who have measured this and seen significant changes in fish behaviour in turbid waters. Just funnelling sediments directly onto the branches creates an unnatural set up straight away.
4. Highest sediment load conditions are typically found on inshore reefs - which are also characterised by lower branching coral cover and higher massive and encrusting corals. So even if there was a real positive influence of damsel fish on branching corals, is it ecologically relevant for these reefs that are largely composed of different coral morphologies? The authors make a nod to this point in the discussion, saying that other forms of corals maybe more susceptible to sedimentation (which I agree with) - but do you see the same interactions with fish with these morphologies - unlikely!

Given these main issues, I am not sure that the results provide a valuable insight into what actually happens on a reef during a sedimentation event. As stated previously, the analysis is sound, just the experiment setup lacks an understanding of sedimentary process on reefs and how this effects reef ecological interactions. At this stage it is difficult to change this - but if the authors can at least adjust statements around the justification of the sedimentation rate used, and talk to the points above I would be happier. As such, I recommend the paper for major edits and another review stage.

Other minor points

1. No explanation of how sediment data was collected around the reefs except related to figure S1 - needs a lot more method explanation, and typically you deploy 3 sediment traps in tandem to accommodate the huge variability often seen
2. Figure s2 talks about sediment vacuum suction - how was that done exactly?
3. Photos were taken above the coral colonies to calculate tissue mortality - how do you do this effectively with a 3d branching coral???
4. You mention the impact of crypto-fauna on sediment clearing rates, but what about differences in coral branching structure between colonies - this is likely to also affect the results
5. I have several issues with part of the discussion from line 411 to 423 where is states that these results a likely underestimation of the positive effects of fish.
 - a. Yes it will have greater impacts on other coral morphologies, but you don't see the same relationships with fish
 - b. Corals feed heterotrophically naturally - so this is ecological relevant and will happen in situ as well
 - c. Natural coral populations are not NATURALLY exposed to >400 mg.cm.day. The related reference - corresponds to ex situ experiments not what has actually been measured on a reef using best methods
 - d. Sentence starting 'Finally, sedimentation alone.....etc' is a huge misinterpretation of the references that are then listed and needs to be addressed. Sedimentation will always have a detrimental effect on corals - it may not be sub-lethal to lethal, but even those corals effective at clearing sediments use energy.

Decision letter (RSOS-190966.R0)

03-Sep-2019

Dear Mr Chase:

Manuscript ID RSOS-190966 entitled "Fishes alleviate the impacts of sediments on host corals" which you submitted to Royal Society Open Science, has been reviewed. The comments from reviewers are included at the bottom of this letter.

In view of the criticisms of the reviewers, the manuscript has been rejected in its current form. However, a new manuscript may be submitted which takes into consideration these comments.

Please note that resubmitting your manuscript does not guarantee eventual acceptance, and that your resubmission will be subject to peer review before a decision is made.

Your resubmitted manuscript should be submitted by 02-Mar-2020. If you are unable to submit by this date please contact the Editorial Office.

on behalf of Prof Kevin Padian (Subject Editor)
openscience@royalsociety.org

Editor comments:

Thanks for your attention to the previous comments of reviewers. One reviewer is still fundamentally unsatisfied but did not provide many specifics, so we must assume that the discontent is general. Another reviewer provides some detailed and specific areas of concern. Taken together these comments could justify an outright rejection, but I want to ask whether you can address their major concerns in a revision -- which would have to address all of these adequately because we cannot have another round of review.

In addition, I think your manuscript would be clearer to general audiences if the following points were specified:

1. You call the relationship between the corals and the fishes mutualistic and refer to them as symbionts, but it is not clear what the interactions are. From the comments of reviewers it seems that the fishes eat cryptofauna lodging in niches within the coral skeletons, but this is not

discussed or quantified so it is not clear how effective this behavior is in cleaning the corals of sediment.

2. You say that other mechanisms by which the fishes clean the corals include the fishes "blowing" on them presumably to get at the cryptofauna, and also by swimming past them, presumably inducing a wake that has the same effect. Are these observations any more than anecdotal and can their effectiveness be traced to the different behaviors of different species?

3. There seems to be some discontent with the level of artificial sedimentation: 100 mg per cm² per day is held to be unnaturally high. This seems less an argument with the behavior you're observing (fish cleaning corals) than with the realism of the rate. You could argue that an artificially high rate is better for testing the effectiveness of the fishes, but critics could claim that it is not realistic. Please address this issue. Thanks again for your revisions.

Reviewers' Comments to Author:

Reviewer: 1

Comments to the Author(s)

The manuscript by Chase et al. is a revision of a previously submitted manuscript. The authors have satisfactorily addressed the reviewer's concerns and in my opinion the manuscript is acceptable for publication following a few minor revisions that are listed below.

Line 83. There has been a revision of Symbiodinium systematics. Symbiodinium is now a genus within the family Symbiodiniaceae (LaJeunesse et al. 2018). The authors should use the new systematics and change Symbiodinium in the sentence to read Family Symbiodiniaceae.

Line 383: the authors state "at night when oxygen levels within the inner branches decline" without providing a citation. Please cite Shashar et al. (1993) for oxygen levels in corals at night.

Line 396. The sentence: "These conditions, therefore, maximise sedimentation rates that can lead to the smothering of corals, characteristic of many inshore reefs [31], leading to greater positive interactions between fishes and their coral hosts." This is an odd sentence that does not work. Please re-write. Perhaps re-work the "characteristic of many inshore reefs" somehow differently in the sentence.

Line 400. The sentence: "Removal of symbiont fishes, therefore, will likely lead to decreases in coral health under high sedimentation, similar to bleaching conditions [24]." Liberman et al. (1995) conducted a damselfish removal experiment and demonstrated that damselfish removal led to lower coral growth and reproductive rates. I think this citation would enhance the sentence. Perhaps change to:
Since removal of symbiont fishes lowers coral growth and reproduction rates (Liberman et al. 1995), fish removal will likely detrimentally affect coral health under high sedimentation, similar to bleaching conditions [24].

Figure 1: I like the fact that the authors have pictures of the fish in the figure. Since in the 'no sediment' treatment no sediment was added (a) and no coral mortality occurred (b), the left side of the figure with the 'no sediment' treatment is not needed in the figure and can be stated verbally. The figure looks odd with only half a figure. I suggest removing the 'no sediment' information from the graph and removing the 'sediment added' from the right hand portion of the graphs and stating that information in the legend. By doing so, Figure 1 will be more in line with Figure 2.

Figure 3: There is a mistake in the figure. The species name underneath the picture for *P. moluccensis* is actually *D. aruanus* and not *P. moluccensis*. Given that the rest of the graphs have *P. moluccensis* data appearing first, I assume that the error is in the species names that appear under the bars and not in the order of the bars.

Reviewer: 2

Comments to the Author(s)

The authors have added text to address some of the more superficial comment of the reviews, but the fundamental issue of this work not justifying the conclusions drawn in this manuscript remain. There is still no meaningful ecological relevance to the study as completed, and the issue of interactive effects of cryptofauna and fishes remains - despite caveats added to the text. My conclusions from my initial review still stand. This work appears to be a result of a study conducted to show a predetermined result, and not a careful examination of a hypothesis developed from careful natural observations, or careful reading of the literature.

Reviewer: 3

Comments to the Author(s)

The manuscript entitled 'Fishes alleviate the impacts of sediments on coral hosts' presents an interesting study on how two species of Damsel fish reduce coral tissue mortality and maintain coral health through the removal of sediments from the surface area of a branching coral. The manuscript is well written and easy to follow, and the accompanying data analysis is thorough and well presented. However, I'm struggling to see value of these results given the lack of natural ecological relevance. This relates to the use of 100 mg.cm².day as the sedimentation rate, which the authors claim falls well within realistic values as measured on the reefs. My main points related to the use of this sedimentation rate are:

1. This rate is exceptionally high and rarely recorded on reefs except during big resuspension events or from dredging events.
2. This value have been measured using sediment traps which those working in these field know vastly over estimate sedimentation rates. One of the references that the authors use to justify the 100 mgcm²day - states this exact fact and does not back up their range of 30 to 360 mg cm² day - in fact the very opposite, and then goes on to make the point about lab experiments failing to use realistic values for these types of experiments
3. And even if 100 mg.cm².day sedimentation rate was a thing - to get to these sedimentation values, there would be exceptionally turbid waters for several days - which I suspect would also impair fish behaviour, which has stated at the end of the discussion, are sensitive to environmental change. Plus there are several studies (e.g. Wenger) who have measured this and seen significant changes in fish behaviour in turbid waters. Just funnelling sediments directly onto the branches creates an unnatural set up straight away.
4. Highest sediment load conditions are typically found on inshore reefs - which are also characterised by lower branching coral cover and higher massive and encrusting corals. So even if there was a real positive influence of damsel fish on branching corals, is it ecologically relevant for these reefs that are largely composed of different coral morphologies? The authors make a nod to this point in the discussion, saying that other forms of corals maybe more susceptible to sedimentation (which I agree with) - but do you see the same interactions with fish with these morphologies - unlikely!

Given these main issues, I am not sure that the results provide a valuable insight into what actually happens on a reef during a sedimentation event. As stated previously, the analysis is sound, just the experiment setup lacks an understanding of sedimentary process on reefs and how this effects reef ecological interactions. At this stage it is difficult to change this - but if the authors can at least adjust statements around the justification of the sedimentation rate used, and talk to the points above I would be happier. As such, I recommend the paper for major edits and another review stage.

Other minor points

1. No explanation of how sediment data was collected around the reefs except related to figure S1 - needs a lot more method explanation, and typically you deploy 3 sediment traps in tandem to accommodate the huge variability often seen

2. Figure s2 talks about sediment vacuum suction – how was that done exactly?
3. Photos were taken above the coral colonies to calculate tissue mortality – how do you do this effectively with a 3d branching coral???
4. You mention the impact of crypto-fauna on sediment clearing rates, but what about differences in coral branching structure between colonies – this is likely to also affect the results
5. I have several issues with part of the discussion from line 411 to 423 where it states that these results are a likely underestimation of the positive effects of fish.
 - a. Yes it will have greater impacts on other coral morphologies, but you don't see the same relationships with fish
 - b. Corals feed heterotrophically naturally – so this is ecologically relevant and will happen in situ as well
 - c. Natural coral populations are not NATURALLY exposed to >400 mg.cm.day. The related reference – corresponds to ex situ experiments not what has actually been measured on a reef using best methods
 - d. Sentence starting 'Finally, sedimentation alone.....etc' is a huge misinterpretation of the references that are then listed and needs to be addressed. Sedimentation will always have a detrimental effect on corals – it may not be sub-lethal to lethal, but even those corals effective at clearing sediments use energy.

Author's Response to Decision Letter for (RSOS-190966.R0)

See Appendix A.

RSOS-192074.R0

Review form: Reviewer 3

Is the manuscript scientifically sound in its present form?

Yes

Are the interpretations and conclusions justified by the results?

Yes

Is the language acceptable?

Yes

Do you have any ethical concerns with this paper?

No

Have you any concerns about statistical analyses in this paper?

No

Recommendation?

Major revision is needed (please make suggestions in comments)

Comments to the Author(s)

On the second revision of this manuscript, I was specifically focusing on if the authors had addressed my previous concerns around ecological relevance and impact. As stated previously, the paper is well written and referenced, and the data has been appropriately analysed. The

authors have made some effort to address my concerns, but I still have not convinced that the experimental design was appropriate. Below are some specific comments regarding this:

1. The authors have provided considerably more information around the sediment deposition data collection, which was great to see. However, this raised new issues for me.
 - a. It's stated that the sedimentation rate measured using traps ranged from 0.2 to 198 mg cm² per day. This is a considerable range that would have significant differences in coral response and reef health. Yet this range is simply averaged to give 137 mg per cm² per day, and then used to justify the 100 mg cm² day used in the experiment. It would be good to see a box plot of these values with a median value to fully see the spread of results and ensure no outliers were skewing this.
 - b. The data from the vacuum method (to also assess sediment deposition), which gave values of 0.28 mg per cm² per day (although the authors used a different unit and gave a higher number of 2.8 g m² day, which I hope was not done intentionally to be misleading??), is 2 to 3 orders of magnitude lower than the sediment traps. This suggests that the average of 137 from the traps is being driven by outliers and using an average value does not represent the system well. The vast difference in the method's results also speaks to my previous point about sediment traps over-estimating rates of sedimentation on reefs, and furthers my point that 100 mg cm² day is not ecological relevant.
2. The authors have provided additional references that have used comparable rates of sediments for testing. All this illustrates to me is that there needs to be a better understanding of sediment dynamics more broadly on reefs, and does not provide relevant justification for the experiment design used.
2. The authors have added appropriate statements around that conditions that create 100 mg cm² day are not common and would only occur during large wave driven sediment resuspension events or dredging events. But then you should really test to see if fish behaviour would be the same during these high flow/wave events and high turbidity. Just because a fish in a clear water, low flow tank can move sediments off a branch, would they still be behaving the same under more stressful conditions?
3. The authors state that 'severe sedimentation levels used herein are relatively uncommon' (line 452) – which I completely agree with – but if its uncommon, how helpful is this fish behaviour in reality to coral reefs (if these fish still act the same under high sedimentation events – see point 2)
4. The authors make the additional point that high sedimentation events can also occur under low turbidity conditions. One example of this is given being sediment deposition by parrotfish. OK this can be true on reefs with high parrotfish populations, but the type of sediment being deposited on the corals would be a very different sediment profile than tested here – being largely composed of carbonate and less 'sticky' etc..
5. The authors also mention that a fragment of each colony was taken for testing (chla etc) – but don't indicate how the fragment taken was selected from each colony. Sedimentation impacts typically have localised impacts on the coral tissue (so underneath the sediment), whereas other areas of the coral may be comparatively healthy. How was bias removed during fragment selection to ensure that the fragment taken was representative for the coral colony?

Decision letter (RSOS-192074.R0)

17-Dec-2019

Dear Mr Chase,

The Subject Editor assigned to your paper ("Fishes alleviate the impacts of sediments on host corals") has now received comments from reviewers. We would like you to revise your paper in accordance with the referee and Associate Editor suggestions which can be found below (not including confidential reports to the Editor). Please note this decision does not guarantee eventual acceptance.

Please submit a copy of your revised paper before 09-Jan-2020. Please note that the revision deadline will expire at 00.00am on this date. If we do not hear from you within this time then it will be assumed that the paper has been withdrawn. In exceptional circumstances, extensions may be possible if agreed with the Editorial Office in advance. We do not allow multiple rounds of revision so we urge you to make every effort to fully address all of the comments at this stage. If deemed necessary by the Editors, your manuscript will be sent back to one or more of the original reviewers for assessment. If the original reviewers are not available we may invite new reviewers.

When submitting your revised manuscript, you must respond to the comments made by the referees and upload a file "Response to Referees" in "Section 6 - File Upload". Please use this to document how you have responded to each of the comments, and the adjustments you have made. In order to expedite the processing of the revised manuscript, please be as specific as possible in your response.

- Ethics statement

- Data accessibility

If you wish to submit your supporting data or code to Dryad (<http://datadryad.org/>), or modify your current submission to dryad, please use the following link:
<http://datadryad.org/submit?journalID=RSOS&manu=RSOS-192074>

- Competing interests

- Authors' contributions

- Acknowledgements

- Funding statement

Kind regards,

Andrew Dunn

on behalf of Prof Kevin Padian (Subject Editor)

Associate Editor Comments to Author:

The reviewer consulted continues to express their concern regarding a number of aspects of your study. Please be careful in providing a revised manuscript and cover letter to make clear how you have addressed the reviewer's concerns in the next version - further rounds of revision will not be possible if you are unable to satisfy the reviewer that the paper should be accepted.

Reviewer comments to Author:

Reviewer: 3

Comments to the Author(s)

On the second revision of this manuscript, I was specifically focusing on if the authors had addressed my previous concerns around ecological relevance and impact. As stated previously, the paper is well written and referenced, and the data has been appropriately analysed. The authors have made some effort to address my concerns, but I still have not convinced that the experimental design was appropriate. Below are some specific comments regarding this:

1. The authors have provided considerably more information around the sediment deposition data collection, which was great to see. However, this raised new issues for me.

a. It's stated that the sedimentation rate measured using traps ranged from 0.2 to 198 mg cm² per day. This is a considerable range that would have significant differences in coral response and reef health. Yet this range is simply averaged to give 137 mg per cm² per day, and then used to justify the 100 mg cm² day used in the experiment. It would be good to see a box plot of these values with a median value to fully see the spread of results and ensure no outliers were skewing this.

b. The data from the vacuum method (to also assess sediment deposition), which gave values of 0.28 mg per cm² per day (although the authors used a different unit and gave a higher number of 2.8 g m² day, which I hope was not done intentionally to be misleading??), is 2 to 3 orders of magnitude lower than the sediment traps. This suggests that the average of 137 from the traps is being driven by outliers and using an average value does not represent the system well. The vast difference in the method's results also speaks to my previous point about sediment traps over-estimating rates of sedimentation on reefs, and furthers my point that 100 mg cm² day is not ecological relevant.

2. The authors have provided additional references that have used comparable rates of sediments for testing. All this illustrates to me is that there needs to be a better understanding of sediment dynamics more broadly on reefs, and does not provide relevant justification for the experiment design used.

2. The authors have added appropriate statements around that conditions that create 100 mg cm² day are not common and would only occur during large wave driven sediment resuspension events or dredging events. But then you should really test to see if fish behaviour would be the same during these high flow/wave events and high turbidity. Just because a fish in a clear water, low flow tank can move sediments off a branch, would they still be behaving the same under more stressful conditions?

3. The authors state that 'severe sedimentation levels used herein are relatively uncommon' (line 452) – which I completely agree with – but if its uncommon, how helpful is this fish behaviour in reality to coral reefs (if these fish still act the same under high sedimentation events – see point 2)

4. The authors make the additional point that high sedimentation events can also occur under low turbidity conditions. One example of this is given being sediment deposition by parrotfish. OK this can be true on reefs with high parrotfish populations, but the type of sediment being deposited on the corals would be a very different sediment profile than tested here – being largely composed of carbonate and less 'sticky' etc..

5. The authors also mention that a fragment of each colony was taken for testing (chla etc) – but don't indicate how the fragment taken was selected from each colony. Sedimentation impacts typically have localised impacts on the coral tissue (so underneath the sediment), whereas other areas of the coral may be comparatively healthy. How was bias removed during fragment selection to ensure that the fragment taken was representative for the coral colony?

Author's Response to Decision Letter for (RSOS-192074.R0)

See Appendix B.

RSOS-192074.R1 (Revision)

Review form: Reviewer 3

Is the manuscript scientifically sound in its present form?

Yes

Are the interpretations and conclusions justified by the results?

Yes

Is the language acceptable?

Yes

Do you have any ethical concerns with this paper?

No

Have you any concerns about statistical analyses in this paper?

No

Recommendation?

Accept as is

Comments to the Author(s)

After a third revision of the manuscript, I am happy to support its publication. The authors have added additional text that directly deals with my previous comments – specifically around 1) the sedimentation rates used, and 2) the ecological relevance of this fish behaviour for coral reefs in general. The authors have clearly stated that this is a first pass at observing the potential role that damselfish may have in alleviating sediment stress for A COLONY as opposed to a reef. Further, the additional clarification on the sediment data gave me a better picture of what was happening on the study site reefs. I don't doubt the paper's findings (given the robust analysis of the data) but I do still have reservations around the actual impact of these mutualistic relationships on long-term coral health and how it relates to reef health, particularly as we don't really understand how these fishes change their behaviour in highly stressful sediment driven conditions. But this is certainly an area for further research.

Decision letter (RSOS-192074.R1)

24-Feb-2020

Dear Mr Chase,

It is a pleasure to accept your manuscript entitled "Fishes alleviate the impacts of sediments on host corals" in its current form for publication in Royal Society Open Science. The comments of the reviewer(s) who reviewed your manuscript are included at the foot of this letter.

Kind regards,

Anita Kristiansen
Editorial Coordinator

on behalf of Kevin Padian (Subject Editor)
openscience@royalsociety.org

Reviewer comments to Author:
Reviewer: 3

Comments to the Author(s)

After a third revision of the manuscript, I am happy to support its publication. The authors have added additional text that directly deals with my previous comments – specifically around 1) the sedimentation rates used, and 2) the ecological relevance of this fish behaviour for coral reefs in general. The authors have clearly stated that this is a first pass at observing the potential role that damsel fish may have in alleviating sediment stress for A COLONY as opposed to a reef. Further, the additional clarification on the sediment data gave me a better picture of what was happening on the study site reefs. I don't doubt the paper's findings (given the robust analysis of the data) but I do still have reservations around the actual impact of these mutualistic relationships on long-term coral health and how it relates to reef health, particularly as we don't really understand how these fishes change their behaviour in highly stressful sediment driven conditions. But this is certainly an area for further research.

Follow Royal Society Publishing on Twitter: [@RSocPublishing](https://twitter.com/RSocPublishing)

Appendix A

Tory J Chase
College of Science and Engineering
Marine and Aquaculture Group
James Cook University
Telephone: +1 603 7759083
E-mail: tory.chase@my.jcu.edu.au
27th of November 2019

Professor Kevin Padian
Subject Editor
Royal Society Open Science
By Electronic Submission

Dear Professor Padian and the Editorial Board for *Royal Society Open Science*,

Thank you for the opportunity to submit a revision of our manuscript “Damselishes alleviate the impacts of sediments on host corals,” for publication as a Research Article in the journal *Royal Society Open Science*. While Reviewers of our manuscript stated the manuscript has the potential to be published in *Royal Society Open Science*, they favored a revised version. The Reviewers suggested including additional details regarding the *in situ* sediment sampling methodology, providing ecological justification of the artificial sedimentation methodology, elaborating the mutualism between coral-dwelling damsselfishes and corals with respect to in colony cryptofauna, and some additional minor editorial revisions.

Please find our responses to Reviewer comments tabulated below. Additionally, changes have been tracked on the updated version of our manuscript for your review. Please note that the line numbers presented at the end of each response refer to the clean and revised version of the manuscript, for easier navigation.

ROYAL SOCIETY OPEN SCIENCE REVIEWER COMMENT	OUR RESPONSE
Academic Editor –	
Thanks for your attention to the previous comments of reviewers. One reviewer is still fundamentally unsatisfied but did not provide many specifics, so we must assume that the discontent is general. Another reviewer provides some detailed and specific areas of concern. Taken together these comments could justify an outright rejection, but I want to ask whether you can address their major concerns in a revision -- which would have to address all of these adequately because we cannot have another round of review.	We have revised the manuscript to address the Reviewers’ comments and thank you for the positive comments. We feel that we are able to more than adequately address all of the major concerns in our revised manuscript.
In addition, I think your manuscript would be clearer to general audiences if the following points were specified: 1. You call the relationship between the corals and the fishes mutualistic and refer to them as symbionts, but it is not clear what	We have clarified the role of cryptofauna in the host corals and have avoided any mention of these damsselfish species eating the cryptofauna, something we have not witnessed or found reference to in the published literature. The mutualistic role between damsselfishes and their host coral are

the interactions are. From the comments of reviewers it seems that the fishes eat cryptofauna lodging in niches within the coral skeletons, but this is not discussed or quantified so it is not clear how effective this behavior is in cleaning the corals of sediment.	now more explicitly explained in the introduction and discussion.
2. You say that other mechanisms by which the fishes clean the corals include the fishes "blowing" on them presumably to get at the cryptofauna, and also by swimming past them, presumably inducing a wake that has the same effect. Are these observations any more than anecdotal and can their effectiveness be traced to the different behaviors of different species?	We have included the "blowing" behavior as personal observation observed during this experiment. We are unable to speak to whether these behaviors are applicable to both species, however sand removal has been observed in other species (see Moyer 1975; Liberman et al. 2005). Branconi et al. 2019 has been included to provide reference to some of these behaviors. Chase et al. (in press) also documents some of the species-specific behaviors (i.e. swimming by, colony visits) which may be connected with cleaning sediment off corals (lines 381-385).
3. There seems to be some discontent with the level of artificial sedimentation: 100 mg per cm² per day is held to be unnaturally high. This seems less an argument with the behavior you're observing (fish cleaning corals) than with the realism of the rate. You could argue that an artificially high rate is better for testing the effectiveness of the fishes, but critics could claim that it is not realistic. Please address this issue. Thanks again for your revisions.	In the revised manuscript we justify our level of artificial sedimentation by; a) directly comparing it with other ex situ experimental studies using similar ranges (allowing for direct comparison and reference), b) details of sediment collection in the field, and c) tailoring our results and statements to reflect that the artificial sediment use more accurately reflect high sedimentation events such as storm events and other natural re-suspension events. We further acknowledge some of the caveats of our current study in the limitations section of the Discussion (lines 432-456).
REVIEWER #1	
The manuscript by Chase et al. is a revision of a previously submitted manuscript. The authors have satisfactorily addressed the reviewer's concerns and in my opinion the manuscript is acceptable for publication following a few minor revisions that are listed below.	We thank the Royal Society Open Science reviewer for the detailed comments.
Line 83. There has been a revision of Symbiodinium systematics. Symbiodinium is now a genus within the family Symbiodiniaceae (LaJeunesse et al. 2018). The authors should use the new systematics and change Symbiodinium in the sentence to read Family Symbiodiniaceae.	Changed as suggested. Symbiodinium has been changed to Symbiodiniaceae spp. With the LaJeunesse et al. 2018 reference (lines 81).

Line 383: the authors state “at night when oxygen levels within the inner branches decline” without providing a citation. Please cite Shashar et al. (1993) for oxygen levels in corals at night.	Changed as suggested (line 398).
Line 396. The sentence: “These conditions, therefore, maximise sedimentation rates that can lead to the smothering of corals, characteristic of many inshore reefs [31], leading to greater positive interactions between fishes and their coral hosts.” This is an odd sentence that does not work. Please re-write. Perhaps re-work the “characteristic of many inshore reefs” somehow differently in the sentence.	We have revised the text to “...suspended in the water column [37,55], maximizing sediment rates that can lead to smothering of corals, a common phenomenon on many inshore reefs [31]...” (lines 410-411).
Line 400. The sentence: “Removal of symbiont fishes, therefore, will likely lead to decreases in coral health under high sedimentation, similar to bleaching conditions [24].” Liberman et al. (1995) conducted a damselfish removal experiment and demonstrated that damselfish removal led to lower coral growth and reproductive rates. I think this citation would enhance the sentence. Perhaps change to: Since removal of symbiont fishes lowers coral growth and reproduction rates (Liberman et al. 1995), fish removal will likely detrimentally affect coral health under high sedimentation, similar to bleaching conditions [24].	Changed as suggested. We have amended the sentence accordingly (lines 416-418).
Figure 1: I like the fact that the authors have pictures of the fish in the figure. Since in the ‘no sediment’ treatment no sediment was added (a) and no coral mortality occurred (b), the left side of the figure with the ‘no sediment’ treatment is not needed in the figure and can be stated verbally. The figure looks odd with only half a figure. I suggest removing the ‘no sediment’ information from the graph and removing the ‘sediment added’ from the right-hand portion of the graphs and stating that information in the legend. By doing so, Figure 1 will be more in line with Figure 2.	Changed as suggested. We have removed the ‘no sediment’ half of Figure 1a and 2b (left side), included the information regarding the ‘no sediment’ information in the Figure 1 description. We have added additional detail in the results regarding the very low levels of sediment and partial mortality in the results section: “Sediments on non-sediment treatment colonies were very low (<0.29 g) for all three treatments, and were likely a result of residual sediments within the aquarium system” (lines 266-267).
Figure 3: There is a mistake in the figure. The species name underneath the picture for P. moluccensis is actually D. aruanus and not P. moluccensis. Given that the rest of the graphs have P. moluccensis data appearing first, I assume that the error is in the	Changed as suggested. We have changed the order of the bars in this figure to accurately reflect the diurnal and nocturnal positions of P. moluccensis first (left side of the figure) and D. aruanus second (right side of the figure).

species names that appear under the bars and not in the order of the bars.	
REVIEWER #2	
The authors have added text to address some of the more superficial comment of the reviews, but the fundamental issue of this work not justifying the conclusions drawn in this manuscript remain. There is still no meaningful ecological relevance to the study as completed, and the issue of interactive effects of cryptofauna and fishes remains - despite caveats added to the text. My conclusions from my initial review still stand. This work appears to be a result of a study conducted to show a predetermined result, and not a careful examination of a hypothesis developed from careful natural observations, or careful reading of the literature.	We have included additional supplementary data and broader reference to the literature to strengthen our justification of the dose of sediment added as well as the ecological relevance of this study. In regard to the cryptofauna, we re-emphasize that any cryptofauna present within coral colonies were randomly distributed among experimental treatments. This means that the presence of cryptofauna might potentially explain some of the variance among replicate coral colonies, but it would not bias the effects of the sediment or fish presence treatments. Such randomised treatments are widely used in experimental studies to factor out additional sources of variation, whilst maintaining ecological relevance. This study was a rigorous and well-replicated investigation of whether and how fish contribute to sediment removal from coral colonies and was in no way conducted to show pre-determined results.
REVIEWER #3	
The manuscript entitled ‘Fishes alleviate the impacts of sediments on coral hosts’ presents an interesting study on how two species of Damselfish reduce coral tissue mortality and maintain coral health through the removal of sediments from the surface area of a branching coral. The manuscript is well written and easy to follow, and the accompanying data analysis is thorough and well presented. However, I’m struggling to see value of these results given the lack of natural ecological relevance. This relates to the use of 100 mg.cm².day as the sedimentation rate, which the authors claim falls well within realistic values as measured other reefs.	We thank the reviewer for their positive comments on our manuscript. In our revised manuscript, we have strengthened our justification of the dose of sediment added as well as the ecological relevance of this study, through additional supplemental information, referencing similar values and concepts in other publications, and referencing these values to be most relevant during high sediment events (i.e. dredging, post-wet season runoff, or high storm/water movement events). In other words, we have clarified that this sediment regime represents a high sediment stress scenario for corals.
My main points related to the use of this sedimentation rate are: 1. This rate is exceptionally high and rarely recorded on reefs except during big resuspension events or from dredging events.	The 100 mg cm⁻² day⁻¹ value that we have used in our experiment are based on (a) sediment trap data from around the Palm Islands, (b) sediment values used in previous publications (to allow direct comparison with our experiment) looking at the impacts of sediment on coral/crypofauna (Stewart et al. 2006 (62-125 mg cm⁻² day⁻¹); Stewart et al. 2013 (>67 mg cm⁻² day⁻¹;), and (c) sediment

	values used in publications investigating the impacts of sediment on coral tissues. Furthermore, due to the distribution of the sediments in the seawater tank, likely <75% of the added sediment actually came into contact with the host coral. Additional references regarding the levels of sediments used in aquaria and in natural systems have been incorporated (see Duckworth et al. 2017 0.5-235 mg cm⁻² day⁻¹, and Erftemeijer et al. 2012 for a comprehensive table to sediment doses in natural and controlled aquaria experiments). We have further discussed the scope of this level of sedimentation in terms of dredging and large resuspension events in the abstract, Introduction, Methods, and Discussion (lines 93-94, 108-109, 163-180, 432-462).
2. This value have been measured using sediment traps which those working in these field know vastly over estimate sedimentation rates. One of the references that the authors use to justify the 100 mgcm2day – states this exact fact and does not back up their range of 30 to 360 mg cm2 day – in fact the very opposite, and then goes on to make the point about lab experiments failing to use realistic values for these types of experiments	We thank the reviewer for bringing this concern to our attention. Our experimental values were based on a comprehensive search of published sedimentation data on inshore GBR reefs, and on experimental values used in published coral/sediment experiments (in particular Stewart et al. 2006, and Erftemeijer et al. 2012). We also measured sedimentation at our study site and have included some of these data in the revised manuscript. We agree that sediment traps are not an ideal method for measuring sedimentation in all instances. However, the way sediments contact living branching corals in situ, is broadly similar to how sediment is collected in traps in that the inner coral branches likely have little resuspension. Furthermore, by contrast to previous findings, recent evidence (Latrille et al. 2019) has highlighted that sediment traps can actually underestimate sediment accumulation rates (by > 60%) in coral reef systems when compared to natural algal turfs. In the updated manuscript, we have included additional text and references to clarify these points (lines 167-180).
3. And even if 100 mg.cm2.day sedimentation rate was a thing – to get to these sedimentation values, there would be exceptionally turbid waters for several days – which I suspect would also impair fish behaviour, which has stated at the end of the	We agree that certain sedimentation events would also increase seawater turbidity. However, from our field sites, often those sites with the highest sedimentation levels (especially coarse grain sediments) had lower turbidity levels. This is because fine

discussion, are sensitive to environmental change. Plus, there are several studies (e.g. Wenger) who have measured this and seen significant changes in fish behaviour in turbid waters. Just funneling sediments directly onto the branches creates an unnatural set up straight away.	sediments suspended in the water column is just one of a range of processes by which sediments can be deposited on corals. For example, the deposition of sediments by parrotfishes is independent of water turbidity but is a major, but often overlooked component of reef sediment dynamics. Indeed, on average, parrotfishes in back reefs habitats on inner-shelf reefs of the GBR deposit approximately 40 kg of sediment m⁻² year⁻¹ (Hoey and Bellwood 2008) which appears to play major roles in sediment dynamics in these systems (Tebbett et al. 2017). Furthermore, as Latrille et al. 2019 suggest, this role of parrotfishes may not be accurately quantified by methods such as sediment traps which can extend into the water column above the point that parrotfishes release sediments onto the reef. We have made additional note of the impact of suspended sediment/turbidity on fish behavior in the limitations section of our Discussion: “It should also be highlighted, that in some circumstances high levels of sediment deposition on reefs are often associated with increased turbidity levels [57], and such high turbidity levels can also alter fish behaviour [90,91]. For example, reduced foraging distances...” (lines 432-444).
4. Highest sediment load conditions are typically found on inshore reefs – which are also characterised by lower branching coral cover and higher massive and encrusting corals. So even if there was a real positive influence of damselfish on branching corals, is it ecologically relevant for these reefs that are largely composed of different coral morphologies? The authors make a nod to this point in the discussion, saying that other forms of corals maybe more susceptible to sedimentation (which I agree with)– but do you see the same interactions with fish with these morphologies – unlikely!	Branching corals are abundant at all of the sites around the Palm Islands, and some studies show that branching Acropora species can even decrease in abundance with distance from shore (Done 1982; Moustaka et al. 2018). The presence of other coral morphologies on inshore reefs does not detract from the ecological relevance of our study.
Given these main issues, I am not sure that the results provide a valuable insight into what actually happens on a reef during a sedimentation event. As stated previously, the analysis is sound, just the experiment setup lacks an understanding of sedimentary	We have incorporated all of the reviewer’s comments into the revised manuscript and attempted to explain the rationale for the sediment methodology and ecological relevance.

process on reefs and how this effects reef ecological interactions. At this stage it is difficult to change this – but if the authors can at least adjust statements around the justification of the sedimentation rate used, and talk to the points above I would be happier. As such, I recommend the paper for major edits and another review stage.	
Other minor points 1. No explanation of how sediment data was collected around the reefs except related to figure S1 – needs a lot more method explanation, and typically you deploy 3 sediment traps in tandem to accommodate the huge variability often seen needs to be addressed. Sedimentation will always have a detrimental effect on corals – it may not be sub-lethal to lethal, but even those corals effective at clearing sediments use energy.	We have added explanation of how the sediment data was collected around the Palm Island reefs in the methods as well as extended methodology in the revised electronic supplementary material, text S1. Notes of potential sub-lethal impacts and those not-detected in our tissue analysis have been included in the Discussion: “...additional branching morphologies/taxa, and other non-visible and sub-lethal impacts. ...” (lines 454-456).
2. Figure s2 talks about sediment vacuum suction – how was that done exactly?	Full details of the underwater sediment vacuum apparatus and collection has been included in the updated electronic supplementary material, text S1.
3. Photos were taken above the coral colonies to calculate tissue mortality – how do you do this effectively with a 3d branching coral???	Partial colony mortality was measured using the same methodology as Stewart et al. 2006 and 2013. We are careful to report only the partial mortality on the two-dimensional surface area of the corals, were we approximate >80% of the mortality occurred.
4. You mention the impact of crypto-fauna on sediment clearing rates, but what about differences in coral branching structure between colonies – this is likely to also affect the results.	P. damicornis colonies (with natural cryptofauna) were randomly allocated to treatments. Therefore, any small (random) variations in branching structure would not lead to significant differences among treatments.
5. I have several issues with part of the discussion from line 411 to 423 where is states that these results a likely underestimation of the positive effects of fish. a. Yes it will have greater impacts on other coral morphologies, but you don’t see the same relationships with fish b. Corals feed heterotrophically naturally – so this is ecological relevant and will happen in situ as well c. Natural coral populations are not NATURALLY exposed to >400 mg.cm.day. The related reference – corresponds to ex situ experiments not what has actually been measured on a reef using best methods	The text in the Discussion has been revised to consider the abundance of branching vs less complex morphologies on inshore reefs as well as how the nature of the fish-coral relationship will change with different coral morphologies. We mention this caveat that different coral morphologies often host less fish and have a flat shape, inferring the impacts of sediment and lack of fish to be more detrimental to the coral host (lines 452-456). We have removed mention of sediments exposed to natural coral populations or diet supplementation.

d. Sentence starting ‘Finally, sedimentation alone.....etc’ is a huge misinterpretation of the references that are then listed an	We have amended the sentence regarding sediment levels natural coral populations are exposed to “...The sediment levels used in the present study were designed to reflect high concentration and prolonged sediment deposition, such as would be experienced during storms and resuspension events rather than average background sedimentation levels [56,57]. However, prior research has documented that select natural coral populations experience sediment deposition rates exceeding 200 mg cm⁻² day⁻¹ [33]...” (lines 445-456). To prevent misinterpretation we have scaled back text regarding underestimation of sediment impacts, the sub-lethal or lethal impacts of sediment on coral, and identify this as potential future research direction (lines 454-457).
--	--

Sincerely,

Tory Chase

Tory J Chase, *sent electronically and on behalf of all co-authors*

E: tory.chase@my.jcu.edu.au

<http://www.coralcoe.org.au/person/tory-chase>

Appendix B

Tory J Chase
College of Science and Engineering
Marine and Aquaculture Group
James Cook University
Telephone: +1 603 7759083
E-mail: tory.chase@my.jcu.edu.au
29th of January 2020

Professor Kevin Padian
Subject Editor
Royal Society Open Science
By Electronic Submission

Dear Professor Padian, Andrew Dunn, and the Editorial Board for *Royal Society Open Science*,

Thank you for the opportunity to submit a revision of our manuscript “Damselishes alleviate the impacts of sediments on host corals,” for publication as a Research Article in the journal *Royal Society Open Science*. While the Reviewers of our manuscript stated the manuscript has the potential to be published in *Royal Society Open Science*, they favored a revised version. The Reviewers suggested including additional details regarding the ecological justification of the artificial sedimentation methodology and some additional minor editorial revisions.

Please find our responses to Reviewer comments tabulated below. Additionally, changes have been tracked on the updated version of our manuscript for your review. Please note that the line numbers presented at the end of each response refer to the clean and revised version of the manuscript.

ROYAL SOCIETY OPEN SCIENCE REVIEWER COMMENT	OUR RESPONSE
Academic Editor –	
The Subject Editor assigned to your paper ("Fishes alleviate the impacts of sediments on host corals") has now received comments from reviewers. We would like you to revise your paper in accordance with the referee and Associate Editor suggestions. The reviewer consulted continues to express their concern regarding a number of aspects of your study. Please be careful in providing a revised manuscript and cover letter to make clear how you have addressed the reviewer's concerns in the next version.	We have revised the manuscript to address the Reviewers' comments. Specifically, to address the ecological relevance of the sediment amount used in this study, we have provided extensive references supporting the validity of these sedimentation values, field data at our study site, and cited experimental studies using similar values for direct comparison. We have also widely expanded the discussion, to put our study into the appropriate context and acknowledge any potential limitations. For instance, we have made it very clear that the sediment deposition levels used are severe, and we have an extensive caveats section in the discussion.
In addition to addressing all of the reviewers' and editor's comments please also ensure that your revised manuscript contains the following sections before the reference list: Ethics statement, Data accessibility, Competing interests, Author's contributions, Acknowledgements, and Funding statement.	These sections are now included in both the updated manuscript and author supplied statements through the submission portal (lines 517-538).

REVIEWER #3	
On the second revision of this manuscript, I was specifically focusing on if the authors had addressed my previous concerns around ecological relevance and impact. As stated previously, the paper is well written and referenced, and the data has been appropriately analysed. The authors have made some effort to address my concerns, but I still have not convinced that the experimental design was appropriate. Below are some specific comments regarding this:	We thank the Royal Society Open Science reviewer for the detailed comments. We now further justify our experimental design through additional sediment trap data discussion, revision of sedimentation rates, elaborating the appropriate context of our study. For example on lines 167-170, we now state that “In addition, a level of 100 mg cm⁻² day⁻¹ was chosen to facilitate comparison with previous research that has explored the impacts of sediment deposition on corals under deposition rates ranging from 0.5 – 600 mg cm⁻² day⁻¹ in natural and controlled ex situ aquaria conditions [13,33,52–54]”.
1. The authors have provided considerably more information around the sediment deposition data collection, which was great to see. However, this raised new issues for me. a. It's stated that the sedimentation rate measured using traps ranged from 0.2 to 198 mg cm² per day. This is a considerable range that would have significant differences in coral response and reef health. Yet this range is simply averaged to give 137 mg per cm² per day, and then used to justify the 100 mg cm² day used in the experiment. It would be good to see a box plot of these values with a median value to fully see the spread of results and ensure no outliers were skewing this. b. The data from the vacuum method (to also assess sediment deposition), which gave values of 0.28 mg per cm² per day (although the authors used a different unit and gave a higher number of 2.8 g m² day, which I hope was not done intentionally to be misleading??), is 2 to 3 orders of magnitude lower than the sediment traps. This suggests that the average of 137 from the traps is being driven by outliers and using	We reiterate that the sediment trap data is a key line of evidence (references of recorded sedimentation values, direct comparison with other experimental studies, in situ field results etc.) to justify a sedimentation rate of 100 mg cm⁻² day⁻¹. In the electronic supplementary material text S1, we have amended the calculations for sediment trap methodology and report that sedimentation rates range from 2-1982 mg cm⁻² day⁻¹ (see Results in electronic supplementary material text S1). We agree that this range is large but suggest that this is a natural occurrence on certain coral reef habitats. In the revised supplement, we now state the average sedimentation mg cm⁻² day⁻¹ per each location (exposed and sheltered), and a figure of the average sedimentation values per sheltered and exposed locations with the corresponding grain size's observed (electronic supplementary material figure s3) to allow the reader to understand both the sediment grain size and spread of the data (electronic supplementary material text S1). We initially recorded the vacuum sediment deposition data as g per m², to be congruent with previous studies reporting sediment deposition with the vacuum method (Latrille et al. 2019, Tebbett et al. 2017). We previously mentioned “all sediment samples were converted into g m⁻² day⁻² for consistency” and was not intended to be misleading. In this updated manuscript, we have amended our calculations and Table S1 in the electronic supplementary material

an average value does not represent the system well. The vast difference in the method's results also speaks to my previous point about sediment traps over-estimating rates of sedimentation on reefs, and furthers my point that 100 mg cm² day is not ecological relevant.	reports the average sediment as mg cm⁻² day⁻¹. Furthermore, we acknowledge our limitations regarding the vacuum sedimentation methodology (see electronic supplementary material text S1). Again, we reference Latrille et al. 2019 that demonstrated a under estimation of sediment deposition rates using sediment traps, compared with algae turfs (lines 473-484).
2. The authors have provided additional references that have used comparable rates of sediments for testing. All this illustrates to me is that there needs to be a better understanding of sediment dynamics more broadly on reefs, and does not provide relevant justification for the experiment design used.	We have selected our experimental sedimentation values to be directly comparable to previous experimental studies (Stewart et al. 2006; 2013, Erfteimeijer et al. 2012) and studies reporting sedimentation on coral reefs (Duckworth et al. 2017). We agree that there needs to be a better understanding of sediment dynamics on reefs, and we have now included an entirely new paragraph in the discussion to highlight this point (lines 473-484). However, the scope of this study does not include examination of broad scale sediment dynamics. As such, we had to work with the levels published in previous studies when deciding on a sediment level to use in the current study.
3. The authors have added appropriate statements around that conditions that create 100 mg cm² day are not common and would only occur during large wave driven sediment resuspension events or dredging events. But then you should really test to see if fish behaviour would be the same during these high flow/wave events and high turbidity. Just because a fish in a clear water, low flow tank can move sediments off a branch, would they still be behaving the same under more stressful conditions?	As stated in this manuscript, this study was conducted to take a step forward towards understanding some fish-coral interaction in more detail and by no means represents an exhaustive examination of every potential environmental condition. We maintain our appropriate justification for the sedimentation levels used in this experiment based on in situ field quantification, direct comparison with previously published studies, and reported sedimentation levels from other experimental and field-based investigations. We have expanded upon how fish behaviour may change and its impact on coral colonies under high water flow and high turbidity events (Johansen et al. 2008; Wenger et al. 2012; 2013; Chase et al. 2020) and made the limitations of this study explicit in the discussion (lines 440-459, 460-484). While behavioural changes are likely to be fish species-specific, our study presents key evidence to show that there is the potential for some behavioural changes to help corals under more stressful conditions (see lines 437-459). However, we have also highlighted

	how sediment deposition on to corals (particularly by parrotfishes) is not always associated with high turbidity or high flow (lines 450-459).
4. The authors state that ‘severe sedimentation levels used herein are relatively uncommon’ (line 452) – which I completely agree with – but if its uncommon, how helpful is this fish behaviour in reality to coral reefs (if these fish still act the same under high sedimentation events – see point 2)	Whether damselfishes act the same under high sedimentation events was not within the scope of this study. We are careful to acknowledge this in the manuscript, and we also broadly discuss how additional environmental changes such as increased water flow or increased turbidity would change fishes’ behaviours (Holbrook et al. 2008; Wenger et al. 2012; 2013; Chase et al. 2020). Furthermore, we also do not claim that this fish behavior is important for coral reefs as a whole. However, it may be important for specific coral colonies that have been exposed to a severe sediment deposition event. We have now clarified the text to ensure that the reader can not misinterpret our findings as applying to entire coral reefs, but instead should be interpreted at the level of individual coral colonies. Our study has shown that if a coral colony hosting D. arunaus is exposed to a severe sediment deposition event than these fish can help ameliorate the impacts of such sediment deposition on that colony.
5. The authors make the additional point that high sedimentation events can also occur under low turbidity conditions. One example of this is given being sediment deposition by parrotfish. OK this can be true on reefs with high parrotfish populations, but the type of sediment being deposited on the corals would be a very different sediment profile than tested here – being largely composed of carbonate and less ‘sticky’ etc..	On most inner-shelf reefs of the GBR parrotfishes are highly abundant, especially Scarus rivulatus, and contribute substantially to sediment dynamics in these locations. Furthermore, the grain size distribution of sediment used in the current study and found in sediment traps (see electronic supplementary material figure S3 and table S3) was based on ‘settled inshore sediments around the Palm Islands’ and included ‘silicate, carbonate, and organic particulates’. This is therefore likely to directly reflect the sediment the majority of parrotfishes deposit on corals in these areas. This is because the majority of parrotfishes on GBR inner-shelf reefs are ‘scrapers’ that rework already settled sediments i.e. they only consume and deposit sediment already settled on the substratum and bound within algal turfs. You are indeed correct that there may be higher levels of carbonate in sediment produced by excavating parrotfishes that actively bioerode the reef. However, these parrotfishes,

	especially Bolbometopon muricatum, dominate outer-shelf reefs where the process of bioerosion is highest. For more details please refer to: Hoey and Bellwood 2008 Tebbett et al. 2017; Bellwood 1996; Bellwood 1995. These details have now been clarified in the manuscript please see lines (450-459).
6. The authors also mention that a fragment of each colony was taken for testing (chla etc) – but don't indicate how the fragment taken was selected from each colony. Sedimentation impacts typically have localised impacts on the coral tissue (so underneath the sediment), whereas other areas of the coral may be comparatively healthy. How was bias removed during fragment selection to ensure that the fragment taken was representative for the coral colony?	Fragments from each P. damicornis colony were selected at random from the top planar surface without knowledge of their allocated treatment. We have included this detail in the updated manuscript “One coral fragment per colony, ~5 cm in length, was haphazardly collected from the top planar surface of each colony...” (lines 154-155).

Sincerely,

Tory Chase

Tory J Chase, sent electronically and on behalf of all co-authors

E: tory.chase@my.jcu.edu.au

<http://www.coralcoe.org.au/person/tory-chase>